# A Siamese Transformer with Hierarchical Refinement for Lane Detection

**Zinan Lv**[1]    **Dong Han**[2*]   **Wenzhe Wang**[2†]   **Danny Z. Chen**[3]
[1]Shanghai JiaoTong University    [2]Zhejiang University
[3]University of Notre Dame

## Abstract

Lane detection is an important yet challenging task in autonomous driving systems. Existing lane detection methods mainly rely on finer-scale information to identify key points of lane lines. Since local information in realistic road environments is frequently obscured by other vehicles or affected by poor outdoor lighting conditions, these methods struggle with the regression of such key points. In this paper, we propose a novel Siamese Transformer with hierarchical refinement for lane detection to improve the detection accuracy in complex road environments. Specifically, we propose a high-to-low hierarchical refinement Transformer structure, called LAne TRansformer (LATR), to refine the key points of lane lines, which integrates global semantics information and finer-scale features. Moreover, exploiting the thin and long characteristics of lane lines, we propose a novel Curve-IoU loss to supervise the fit of lane lines. Extensive experiments on three benchmark datasets of lane detection demonstrate that our proposed new method achieves state-of-the-art results with high accuracy and efficiency. Specifically, our method achieves improved F1 scores on the OpenLane dataset, surpassing the current best-performing method by 5.0 points.

## 1   Introduction

Lane detection is a fundamental task in Autonomous Driving Systems (ADS), which enables a vehicle to localize its relative position and avoid potential risks. It plays an important role in many downstream tasks, such as driving route planning, road tracking, and adaptive cruise control. Recently, lane detection methods based on computer vision have attained lots of achievements. Compared to methods that combine GPS/INS (Inertial Navigation System) [2], lane detection methods that incorporate only a camera are cheaper and safer to apply.

Early lane detection research focused on hand-crafted features and applied methods such as Hough Transform [16] or Kalman Filter [36] to filter out unreasonable lanes. However, manually extracted features often fail in complex scenarios. Convolutional Neural Network (CNN) based methods [10, 13, 38, 39] arose to cope with different scenarios, which greatly improved the accuracy of lane detection. These methods often rely on straight-line anchors [12, 15, 31, 32, 35] or parametric curves [18, 33, 9, 4] to detect lane lines. Although CNN-based methods can handle different scenarios, they still struggle with realistic road environments, especially when involving strong light, shadows, or dense traffic. Due to the thin and long characteristics of lane lines, lane detection requires a lot of contextual information, and CNN-based methods may be ineffective and incapable of clustering global semantics information and combining it with finer-scale features.

Recently, attention-based methods (e.g., CLRNet [41]) have shown promising capability in lane detection by describing a lane line as a series of key points. By taking the lane lines as a whole unit,

---

*Corresponding Author. Email: phhandong@outlook.com
†Corresponding Author. Email: wangwnezhe@zju.edu.cn

38th Conference on Neural Information Processing Systems (NeurIPS 2024).

these methods can make use of global semantics information and finer-scale features, which help the detection of lane lines in complex environments. However, since these methods mainly rely on finer-scale information to identify the position of each key point, their detection results may have large deviations when there is local occlusion or blurring. In addition, former studies did not take into account the thin and long structure of lane lines during supervision, thus often leading to inaccurate detection when the lane line has a certain curvature.

In this paper, we propose a novel Siamese Transformer with hierarchical refinement for lane detection to improve the detection accuracy in realistic road environments, especially when roads are obscured by other vehicles or affected by poor outdoor lighting conditions. Two attributes of our proposed method contribute to its universality for the lane detection task. First, to address the under utilization of finer-scale information when fine-tuning the positions of key points, we propose a novel Siamese Transformer structure with shared parameters, called LAne TRansformer (LATR), which can integrate global semantics information and finer-scale features. Simultaneously, we develop a high-to-low hierarchical refinement scheme to refine the key points of lane lines so that the network can fully learn information at different scales. Second, to take the thin and long structure of lane lines into account, we propose a novel Intersection over Union (IoU) loss called Curve-IoU (CIoU) for lane detection. Compared to the common IoU loss, we supervise the fit of lane lines at different locations for the thin and long structure of the lane lines, which helps accurately detect lane lines with curves.

We evaluate our proposed method on three benchmark datasets for lane detection, OpenLane [5], CULane [26], and Tusimple[34], and achieve state-of-the-art results. Our main contributions can be summarized as follows:

- We propose a novel Siamese Transformer with hierarchical refinement for lane detection to improve the detection accuracy in realistic road environments, especially when roads are obscured by other vehicles or affected by poor outdoor lighting conditions.

- We propose a high-to-low hierarchical refinement Transformer structure called LATR to refine the key points of lane lines so that the network can fully integrate global semantics information and finer-scale features.

- Exploiting the thin and long structure of lane lines, we propose a novel Curve-IoU loss to supervise the fit of lane lines at different locations, which helps the regression of the curves.

- We achieve state-of-the-art results on three benchmark datasets, with 5.0% improvement in F1 score compared to the best-known method on the OpenLane dataset.

## 2 Related Work

Early lane detection studies relied on hand-crafted features [7, 23]. Due to their limited feature capturing capability and low robustness, these methods often fail in complex conditions.

To improve the robustness of lane detection under different environments, segmentation-based methods [10, 13, 38, 39] were introduced to lane detection. These methods typically apply post-processing operations such as curve fitting and clustering on pixel-level segmentation maps to generate final results. Compared to traditional methods, segmentation-based methods are able to capture more plentiful visual features and spatial structure information, thus achieving better performance than traditional detection methods. However, per-pixel-based segmentation methods incur high computational costs, have limited real-time capability, and struggle with learning lane line specific long and thin characteristics.

To address these issues, LaneNet [25] introduced a branched, multi-task architecture to cast the lane detection task as an instance segmentation problem. This method is more robust to variations in road conditions compared to the previous methods, but it is more time-consuming. RESA [40] proposed to aggregate spatial information by shifting sliced feature maps, which obtains good real-time results but still fails under complex road conditions. Furthermore, the output lane lines of most of the above methods may not be continuous.

To attain more continuous lane lines with higher efficiency, recently curve-based methods [18, 9, 4, 33] viewed the lane detection task as a polynomial regression problem and utilized parametric curves to fit lane lines. These methods depend heavily on the parameters of the curves (e.g., $x = ay^3 + by^2 + cy + d$, where $(x, y)$ denotes the coordinates of a lane line pixel and $a, b, c$, and $d$ are the parameters of a

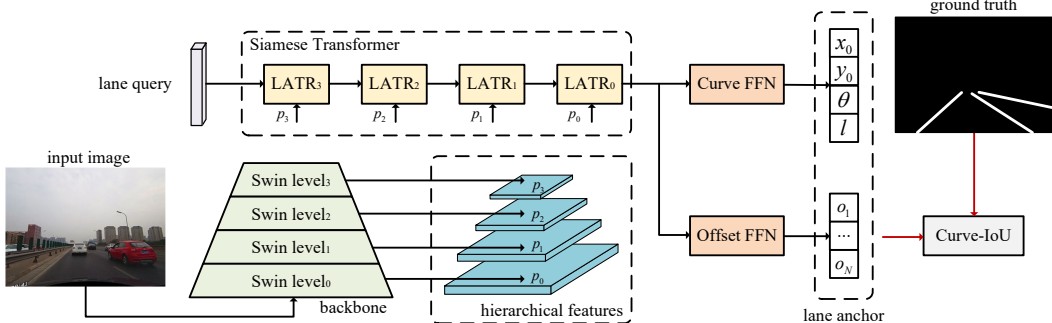

Figure 1: The overall architecture of our proposed method. It includes a multi-scale backbone to extract hierarchical features from the input image, a Siamese Transformer structure named LATR to integrate global semantics information and finer-scale features, and a Curve-IoU loss to supervise the fit of lane lines for model training. Swin level 1-3 denotes the multi-scale backbone using the Swin Transformer. FFN represents a feed-forward network.

curve function). PloyLaneNet [33] first proposed an end-to-end deep polynomial regression method that directly outputs the parameters. To improve the stability and efficiency, BézierLaneNet [9] proposed a parametric Bézier curve to model the geometric shape of lane lines. However, because of the limited learning ability of global information, the accuracy of these curve-based methods is not satisfactory on large datasets, especially in complex road conditions.

Attention-based methods [32, 18] were introduced to the computer vision field, and were proven to be able to capture long-range information. LaneATT [32] introduced an attention mechanism to anchor-based lane detection methods. Based on PolyLaneNet [33], LSTR [18] was proposed with high inference efficiency but relatively low accuracy, especially in some complex road environments. PriorLane [30] improved the accuracy of prediction compared to LSTR with pre-training and local prior. However, there is still a gap in accuracy between the contemporaneous Transformer-based methods and CNN-based methods, and the reason can be attributed to the shortcut of the multi-head self-attention mechanism which neglects the characteristics of different frequencies [28].

## 3 Method

In this section, we present the proposed Siamese Transformer with hierarchical refinement for lane detection. First, we describe the overall architecture of our proposed network. Then we explain each key component of the proposed network, including the LAne TRansformer (LATR) and the proposed Curve-IoU loss. Finally, we provide the inference details of our network.

### 3.1 Overall Architecture

We present the overall architecture of our proposed method in Fig. 1. It consists of a multi-scale backbone and a Siamese Transformer structure named LATR. An input image $\mathbb{R}^{H \times W \times C}$ is first fed into the backbone to obtain hierarchical features from high to low levels, which are then refined by LATR with the supervision of the Curve-IoU loss. Next, we use two different detection feed-forward networks (FFN) to generate (1) lane line properties including the start point $(x_0, y_0)$, angle of inclination $\theta$, and length $l$ of each lane line, and (2) the offset map $\{o_i\}_{i=1}^{P}$. Finally, key points of the lane line are produced by post-processing, which can be expressed as:

$$x_i = \tan\theta \times (y_i - y_0) + x_0 + o_i, \tag{1}$$

where $i$ denotes the $i$-th key point of the lane line. All the key points are sampled at equal intervals based on the $y$-axis, which can be expressed as $y_i = i\frac{H}{P+1}$, where $H$ and $P$ represent the image height and the number of key points, respectively.

### 3.2 Lane Transformer

**Transformer structure.** Our proposed LATR employs the Siamese Transformer structure based on the hierarchical features extracted by the backbone of the network. Inspired by the recent Transformer-based networks with Siamese structures [1, 14], we develop a high-to-low refinement structure to obtain features of the predicted lane line. The detailed structure of LATR is illustrated in Fig. 2.

Specifically, we denote a lane query as $Q_d = \{q_j\}_{j=1}^N$, where $N$ represents the dimension of the decoder embeddings and $d$ is the number of the scales. The lane query is a series of one-dimensional features of the lane lines, which can be further processed to obtain the key points of the lane lines. The input hierarchical features $P_d \in \mathbb{R}^{H_d \times W_d \times C'}$ are extracted by the multi-scale backbone (we use Swin Transformer [19] for it), and then are sorted from high-level to low-level. Next, we flatten it into $M_d \in \mathbb{R}^{H_d W_d \times C'}$ with 2-D positional embedding.

Different from DETR [3], we integrate the lane query in the Transformer encoder and then employ Deformable Attention [42] as cross-attention in the decoder process. The whole process can be expressed as:

$$k = Softmax(\frac{Q'_{d-1} M'_d{}^T}{\sqrt{z_d}})M'_d + Q'_{d-1},$$

$$Q_d = DA(k, k^T, Q'_{d-1}) + Q_{d-1}, \quad (2)$$

where $Q'_d$ and $M'_d$ represent the lane query and high-to-low hierarchical features after positional embeddings respectively, $DA()$ denotes the Deformable Attention, and $z_d$ represents the sequence length.

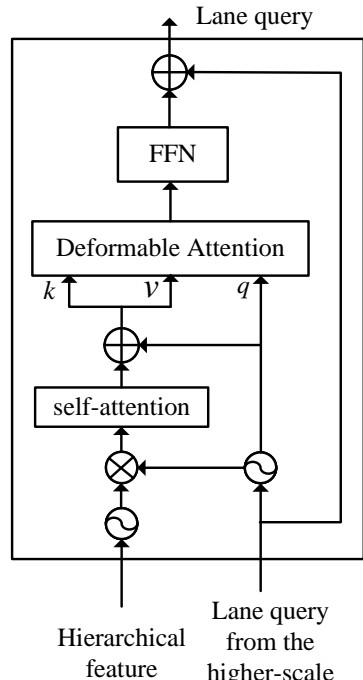

Figure 2: The detailed structure of our proposed LAne TRansformer (LATR). We employ a high-to-low refinement structure in which the input lane query is refined by higher-scale features.

Compared with DETR, our proposed Siamese Transformer integrates the lane query during the encoder. The multi-scale features from the backbone have different shapes. Note that integration during the encoder can unify the decoder's input scales with less information loss. When the scales are unified, the network can be made into a Siamese structure with shared parameters, which leads to a big efficiency improvement.

**High-to-low refinement structure with shared parameters.** Recent studies have shown that Vision Transformer (ViT) is good at extracting low-frequency information (in other words, global semantics information) from images [27]. However, lane lines are characteristically thin and long, and their accurate detection often requires integrating finer-scale features and global semantics information. High-level information can help determine the structural information of lane lines while low-level information can help adapt key points of the lane lines. Therefore, we adopt a high-to-low refinement structure to refine the lane query. Given hierarchical features $P_d \in \mathbb{R}^{H_d \times W_d \times C'}$ from the multi-scale backbone, we first flatten it into $M_d \in \mathbb{R}^{H_d W_d \times C'}$ with 2-D positional embedding. Then we employ cross-attention to integrate $M_d$ with the lane query into $S_d \in \mathbb{R}^{N \times C'}$. Since $S_d$ shares the same shape, the same parameters can be used during refinement between multi-scales. By refining the lane query from high to low, the network can adapt the key points of the lane lines from coarse to fine, which helps the Transformer learn high-frequency information of the image. After the refinement, we employ a detection FFN to generate the lane anchor and finally attain the predicted lane lines with bi-linear interpolation between the key points.

### 3.3 Curve-IoU Loss

We represent a lane line as a sequence of key points, and we need a loss to supervise the regression of lane lines. In previous work, LIoU [41] was proposed as a similarity function to calculate the distance between predicted key points and the ground truth. Although the LIoU loss ensures the scale consistency of the lane shape, it does not match the L1 distance when there is a long distance between the prediction lane line and the ground truth. Fig. 3(a) shows a typical curve scenario that

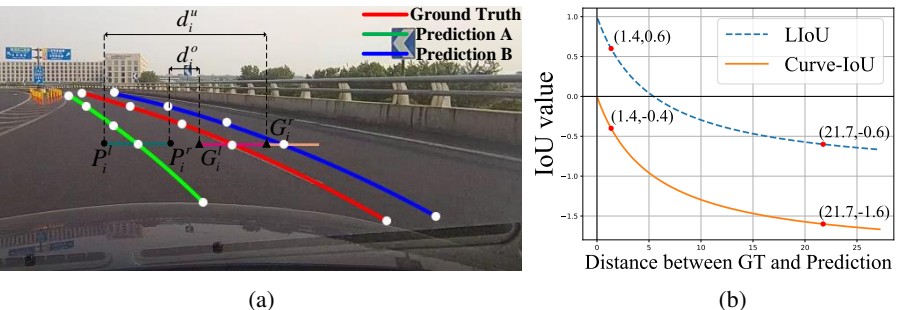

(a)                                    (b)

Figure 3: (a) a typical failure scenario for LIoU, which cannot measure the distances between Ground Truth and Predictions A and B; (b) the values of LIoU and Curve-IoU.

LIoU cannot describe correctly. In such cases, lane lines are inclined and curved, and the LIoU loss is unable to judge which prediction lane line is closer to the ground truth. In Fig. 3, the values for Prediction A and Prediction B are $-0.6$ and $0.6$, respectively. However, the L1 distances for them are $8e$ and $1.5e$, respectively.

To bridge this gap, we propose Curve-IoU (CIoU). Different from the traditional IoU, we add a penalty term to IoU so that the spatial distance between each pair of lane lines will be represented more accurately, especially for those with a long distance between the prediction lane line and the ground truth. We use a sequence of points with a certain width $e$ to present a lane line and calculate the Intersection over Union, which can be written as:

$$d_i^o = \min(P_i^r, G_i^r) - \max(P_i^l, G_i^l),$$
$$d_i^u = \max(P_i^r, G_i^r) - \min(P_i^l, G_i^l),$$
(3)

where $P_i^l = x_i - e$ and $P_i^r = x_i + e$, and $G_i^l$ and $G_i^r$ are defined similarly. $IoU$ can be calculated as $IoU = \frac{d_i^o}{d_i^u}$. Then we add the L1 distance between each corresponding pair of lines as the penalty term to the overlap ranges, as:

$$\mathcal{L}_{IoU} = 1 - \frac{\sum_{i=j}^{k}(d_i^o - ReLU(d_i^u - 4e))}{\sum_{i=j}^{k} d_i^u},$$
(4)

where $j$ is the first valid point in the lane line. Then the CIoU loss can be calculated as $CIoU = 1 - \mathcal{L}_{IoU}$.

Exploiting the thin and long structures of lane lines, our CIoU loss can supervise the regression of lane lines better. As shown in Fig. 3(b), the values of LIoU and CIoU for Prediction A are $-0.6$ and $-1.6$ respectively, which means that our CIoU can handle these scenarios more precisely.

### 3.4 Inference Details

In order to reduce the training costs, we generate only the lane anchor during training. The loss is calculated by comparing the lane anchor to the ground truth. Our method generates a fixed number of lane anchors, which is much larger than the maximum number of lane lines in the image. Besides the Curve-IoU loss, we also utilize a classification accuracy loss to determine whether the lane anchor is a lane line or the background. The classification accuracy loss can be described as:

$$\ell_{cls} = \sum_{i=1}^{L} \mathcal{L}(p_i, g_i),$$
(5)

where $\mathcal{L}(\cdot, \cdot)$ denotes the cross-entropy loss and $L$ is the number of sequences. If the $i$-th predicted lane line matches the ground truth, then $g_i = 1$ (0 otherwise). The objective function of our network can be represented as:

$$\ell = \lambda_{cls}\ell_{cls} + \lambda_{reg}CIoU,$$
(6)

where $\lambda_{cls}$ and $\lambda_{reg}$ are importance weight values. We supervise each generated lane anchor during training, while during inference we use Non-Maximum Suppression (NMS) [24] to exclude incorrect lane lines. Finally, we utilize bi-linear interpolation to produce the predicted lane lines of the image.

| Method | Backbone | F1 | Up & Down | Curve | Extreme Weather | Night | Intersection | Merge & Split | FPS(↑) | GFlops(↓) |
|---|---|---|---|---|---|---|---|---|---|---|
| LaneATT [32] | ResNet18 | 28.3 | 25.3 | 25.8 | 32.0 | 27.6 | 14.0 | 24.3 | 153 | **9.3** |
| LaneATT [32] | ResNet34 | 31.0 | 28.3 | 27.4 | 34.7 | 30.2 | 17.0 | 26.5 | 129 | 18.0 |
| PersFormer [5] | EfficientNetB7 | 42.0 | 40.7 | 46.3 | 43.7 | 36.1 | 28.9 | 41.2 | - | - |
| CondLaneNet [17] | ResNet18 | 52.3 | 55.3 | 57.5 | 45.8 | 46.6 | 48.4 | 45.5 | **173** | 10.2 |
| CondLaneNet [17] | ResNet34 | 55.0 | 58.5 | 59.4 | 49.2 | 48.6 | 50.7 | 47.8 | 128 | 19.6 |
| CondLaneNet [17] | ResNet101 | 59.1 | 62.1 | 62.9 | 54.7 | 51.0 | 55.7 | 52.3 | 47 | 44.8 |
| CLRNet [41] | ResNet18 | 52.3 | 55.3 | 57.5 | 45.8 | 46.6 | 48.4 | 45.5 | 168 | 11.9 |
| CLRNet [41] | ResNet34 | 55.0 | 58.5 | 59.4 | 49.2 | 48.6 | 50.7 | 47.8 | 124 | 19.6 |
| CLRNet [41] | ResNet101 | 59.1 | 62.1 | 62.9 | 54.7 | 51.0 | 55.7 | 52.3 | 46 | 44.8 |
| CondLSTR [6] | ResNet18 | 63.3 | 58.3 | 64.6 | 55.9 | 53.4 | 56.3 | 66.8 | 105 | 13.7 |
| CondLSTR [6] | ResNet34 | 65.6 | 59.1 | 66.7 | 57.2 | 55.6 | 57.5 | 68.3 | 91 | 23.2 |
| CondLSTR [6] | ResNet101 | 67.8 | 62.2 | 68.0 | 59.8 | 57.4 | 59.1 | 69.4 | 45 | 50.2 |
| **Our method** | tiny | 68.3 | 69.5 | 65.3 | 52.4 | 50.9 | 56.5 | 69.6 | 172 | 10.2 |
| **Our method** | small | 69.1 | 70.3 | 67.0 | 54.5 | 51.7 | 58.5 | 71.3 | 111 | 17.6 |
| **Our method** | base | **69.7** | **70.5** | **69.1** | 55.7 | 52.2 | **59.9** | **71.6** | 69 | 16.8 |

Table 1: Comparison results of recent methods and our method on the OpenLane dataset. In order to compare the computation speeds in the same setting, we remeasure FPS on the same machine with an RTX3090 GPU using open-source code (if code is available). The best results in each column are marked as **bold** and the second best results are underlined.

| Method | Backbone | Total | Normal | Crowded | Hlight | Shadow | Noline | Arrow | Curve | Cross | Night | FPS(↑) | GFlops(↓) |
|---|---|---|---|---|---|---|---|---|---|---|---|---|---|
| SCNN [26] | VGG16 | 71.60 | 90.60 | 69.70 | 58.50 | 66.90 | 43.40 | 84.10 | 64.40 | 1990 | 66.10 | 7.5 | 328.4 |
| UFLD [29] | ResNet18 | 68.40 | 87.70 | 66.00 | 58.40 | 62.80 | 40.20 | 81.00 | 57.90 | 1743 | 62.10 | **341** | 8.4 |
| UFLD [29] | ResNet34 | 72.30 | 90.70 | 70.20 | 59.50 | 69.30 | 44.40 | 85.70 | 69.50 | 2037 | 66.70 | 184 | - |
| LSTR [18] | ResNet18 | 68.72 | 86.78 | 67.34 | 56.63 | 59.82 | 40.10 | 78.66 | 56.64 | 1166 | 59.92 | 126 | **2.9** |
| Laneformer [11] | ResNet18 | 71.71 | 88.60 | 69.02 | 64.07 | 65.02 | 45.00 | 81.55 | 60.46 | 25 | 64.76 | - | - |
| Laneformer [11] | ResNet34 | 74.70 | 90.74 | 72.31 | 69.12 | 71.57 | 47.37 | 85.07 | 65.90 | 26 | 67.77 | - | - |
| Laneformer [11] | ResNet50 | 77.06 | 91.77 | 75.41 | 70.17 | 75.75 | 48.73 | 87.65 | 66.33 | 19 | 71.04 | - | - |
| RESA [40] | ResNet34 | 74.50 | 91.90 | 72.40 | 66.50 | 72.00 | 46.30 | 88.10 | 68.60 | 1896 | 69.80 | 51 | - |
| RESA [40] | ResNet50 | 75.30 | 92.10 | 73.10 | 69.20 | 72.80 | 47.70 | 88.30 | 70.30 | 1503 | 69.90 | 39 | - |
| PriorLane [30] | ResNet18 | 76.27 | 92.36 | 73.86 | 68.26 | 78.13 | 49.60 | 88.59 | 73.94 | 2688 | 70.26 | - | - |
| ADNet [37] | ResNet18 | 77.56 | 91.92 | 75.81 | 69.39 | 76.21 | 51.75 | 87.71 | 68.84 | 1133 | 72.33 | 87 | - |
| ADNet [37] | ResNet34 | 78.94 | 92.90 | 77.45 | 71.71 | 79.11 | 52.89 | 89.90 | 70.64 | 1499 | 74.78 | 77 | - |
| CondLaneNet [17] | ResNet18 | 78.14 | 92.87 | 75.79 | 70.72 | 80.01 | 52.39 | 89.37 | 72.40 | 1364 | 73.23 | 154 | 10.2 |
| CondLaneNet [17] | ResNet101 | 79.48 | 93.47 | 77.44 | 70.93 | 80.91 | 54.13 | 90.16 | 75.21 | 1201 | 74.80 | 45 | 44.8 |
| CLRNet [41] | ResNet18 | 79.58 | 93.30 | 78.33 | 73.71 | 79.66 | 53.14 | 90.25 | 71.56 | 1321 | 75.11 | 152 | 11.9 |
| CLRNet [41] | ResNet101 | 80.13 | 93.85 | 78.78 | 72.49 | 82.33 | 54.48 | 89.79 | 75.57 | 1262 | 75.51 | 68 | 42.9 |
| CLRNet [41] | DLA34 | 80.47 | 93.73 | 79.59 | 75.30 | **82.51** | 54.58 | **90.62** | 74.13 | 1155 | 75.37 | 101 | 18.5 |
| **Our method** | tiny | 80.01 | 93.48 | 79.31 | 73.91 | 82.30 | 53.56 | 89.88 | 69.13 | 1112 | 75.41 | 173 | 10.2 |
| **Our method** | small | 80.43 | 93.65 | 79.76 | 73.34 | 81.21 | 53.56 | 89.76 | 72.58 | **1022** | 75.54 | 121 | 17.6 |
| **Our method** | base | **80.85** | **93.92** | **80.21** | **76.04** | 81.65 | **55.42** | 89.53 | **75.66** | 1043 | **75.81** | 78 | 16.8 |

Table 2: Comparison results of recent methods and our method on the CULane dataset.

# 4 Experiments

## 4.1 Datasets

To demonstrate the effectiveness of our proposed method in realistic road environments, we evaluate it on three benchmark datasets: OpenLane [5], CULane [26], and Tusimple [34]. OpenLane is a real-world large-scale lane detection dataset, which contains 160K and 40K images as the training and validation sets, respectively. The validation set consists of six realistic road scenarios and annotates 14 lane categories (including white dotted line, double yellow solid, left/right curb, and so on). CULane is a widely-used large dataset for lane detection including eight hard-to-detect scenarios in urban areas and on highways, with 88K and 34K images as the training and validation sets, respectively. Tusimple is also a widely-used dataset with images collected on US highways under clear weather, which contains 3K images for training and 2K for validation.

## 4.2 Evaluation Metrics

For OpenLane [5] and CULane [26], we adopt the F1-score measure proposed by SCNN [26] as the evaluation metric. Intersection-over-Union (IoU) between the ground truth (GT) label and each predicted lane line of the model is calculated to determine whether a sample is True Positive (TP), False Positive (FP), or False Negative (FN). The ways to calculate IoU and F1-score can be found in the supplemental material. For Tusimple [34], the evaluation metrics are composed of three official indicators: Accuracy, False Positive Rate (FPR), and False Negative Rate (FNR). The way to calculate Accuracy can be found in the supplemental material.

| Method | Backbone | F1 | Acc | FP(↓) | FN(↓) |
|---|---|---|---|---|---|
| SCNN | VGG16 | 95.97 | 96.53 | 6.17 | **1.80** |
| RESA | ResNet34 | 96.93 | 96.82 | 3.63 | 2.48 |
| UFLD | ResNet18 | 87.87 | 95.82 | 19.05 | 3.92 |
| UFLD | ResNet34 | 88.02 | 95.86 | 18.91 | 3.75 |
| PolyLaneNet | EfficientNetB0 | 90.62 | 93.36 | 9.42 | 9.33 |
| LaneATT | ResNet18 | 96.71 | 95.57 | 3.56 | 3.01 |
| LaneATT | ResNet122 | 96.06 | 96.10 | 5.64 | 2.17 |
| LSTR | ResNet18 | 96.84 | 96.18 | 2.91 | 3.38 |
| ADNet | ResNet18 | 96.90 | 96.23 | 2.91 | 3.29 |
| ADNet | ResNet34 | 97.31 | 96.60 | 2.83 | 2.53 |
| PriorLane | ResNet18 | 97.15 | 96.58 | 3.91 | 2.95 |
| Laneformer | ResNet18 | 97.30 | 96.54 | 4.35 | 2.36 |
| Laneformer | ResNet34 | 97.41 | 96.56 | 5.39 | 3.37 |
| Laneformer | ResNet50 | 97.56 | 96.80 | 5.60 | 1.99 |
| CLRNet | ResNet18 | 97.41 | 96.84 | 2.28 | 1.92 |
| CLRNet | ResNet101 | 97.68 | 96.83 | 2.37 | 2.38 |
| **Our method** | tiny | 97.76 | 96.85 | 2.55 | 2.03 |
| **Our method** | small | **98.01** | 96.79 | 2.13 | 2.17 |
| **Our method** | base | 97.85 | **96.95** | **1.86** | 2.34 |

Table 3: Comparison results on the Tusimple dataset.

### 4.3 Implementation Details

In the experiments, we adopt the Swin Transformer [19] as the pre-trained backbone of our network. We divide the versions of the backbone of the network based on the size of the Swin Transformer into three categories: tiny, small, and big, which are consistent with the work in [19]. For data augmentation, we adopt the affine transformation method (horizontal flip, rotation), brightness, and saturation addition method. All the input images are reshaped into $800 \times 320$ pixels each for both the training and inference stages. For the number of lane anchors in an image, we set it to 150. In the optimization process, we adopt AdamW [21] and the cosine decay learning rate strategy [20] with the initial learning rate set to 6e-4. A batch size of 32 and training epoch numbers of 10, 20, and 90 are used for OpenLane, CULane, and Tusimple, respectively. All the experiments are conducted on a machine with a single NVIDIA RTX3090 GPU with 24GB memory.

### 4.4 Experimental Results

#### 4.4.1 Results on OpenLane

Comparisons of results by recent methods and our method on the OpenLane dataset are shown in Table 1. Our method achieves state-of-the-art results in F1 score. Specifically, our method achieves F1 scores of 68.3, 69.1, and 69.7, surpassing those of the best-known method CondLaneNet by 5.0, 3.5, and 1.9 points, respectively. Further, our method achieves the best performances in four out of six scenarios, showing the robustness of our method. Among these, the "Up & Down" and "Curve" categories are 7.2% and 1.1% higher than the previous best results, respectively. These results demonstrate that the proposed Siamese Transformer with the hierarchical refinement structure can deal with lane lines in realistic road environments very well. This is because our proposed LATR integrates global semantics information and finer-scale features, which help refine key points when roads are obscured by other vehicles or affected by poor outdoor lighting conditions. Simultaneously, the proposed Curve-IoU can supervise the regression of curves well, which helps accurate detection of lane lines.

#### 4.4.2 Results on CULane

We compare the results by recent known methods and our method on the CULane dataset in Table 2. In the total F1 score, our method achieves state-of-the-art results, with 0.38% improvement over the best-known CLRNet. Among all the eight difficult scenarios, our method achieves the best results on five of them. Specifically, for the "Crowded" and "Hlight" scenarios, our method achieves

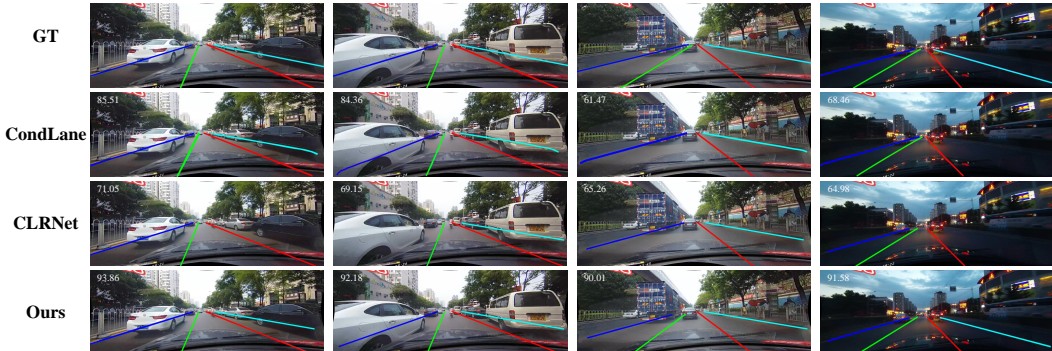

Figure 5: Visualization results of Ground Truth (GT), CondLaneNet [17] (CondLane), CLRNet [41], and our method on CULane [26]. The results of CondLaneNet and CLRNet are generated with ResNet18 and ours are generated with Swin Transformer tiny. Different lane lines are represented by different colors. The F1 score for each predicted image is labeled in the top left corner of the image.

80.21% and 76.04% F1 scores respectively, surpassing those of CLRNet by 0.62% and 0.74% points, respectively. These results demonstrate that our method can deal with realistic road environments, especially when roads are obscured by other vehicles or affected by poor outdoor lighting conditions.

Fig. 5 shows some visual results of several known methods and our method on the CULane dataset. CondLaneNet is an open-source dynamic convolution-based method and CLRNet is the second best-performing method on CULane. The results of CondLaneNet and CLRNet are generated with ResNet18 as the backbone and ours are generated with Swin Transformer tiny as the backbone. The visual results show that when the roads are crowded and lane lines are covered by other vehicles, our method can detect lane lines better and the lane regression is closer to the ground truth. What's more, our method also outperforms these methods in night scenarios, proving that our method can adapt to different lighting environments with good robustness.

### 4.4.3 Results on Tusimple

Comparison results of recent methods and our method on the Tusimple dataset are given in Table 3. This dataset consists of images captured in different weather conditions. Our method achieves state-of-the-art results in F1 score, Accuracy, and False Positive Rate (FP), demonstrating that our method can be adapted to both complex urban environments and simple highway scenarios. Note that because the dataset was collected on US highways where the road conditions are relatively simple (lane features are more obvious and clear), the results of various methods are close.

## 4.5 Ablation Studies

### 4.5.1 LAne TRansformer (LATR)

To fully integrate global semantics information and finer-scale features, we propose a high-to-low hierarchical refinement Transformer structure for lane detection called LAne TRansformer (LATR), which helps identify key points especially when roads are crowded or affected by blurring. Fig. 4 shows some high-to-low attention maps of our LATR. From Fig. 4, one can see that attention is gradually focused on key points on both sides of the road from high to low. High-level attention is extended along the road from the near end to the far end of the road, initially defining the overall structure of the lane lines. Low-level attention focuses on finer-scale features of the lane lines, which refine the key points of the lane lines from higher-levels to lower-levels.

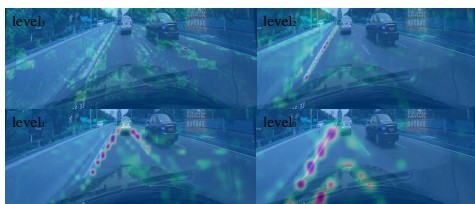

Figure 4: High-to-low attention maps of our proposed LATR.

We compare the F1 scores of our Lane Transformer (LATR) and existing Transformer structures with the same backbone Swin Transformer tiny on CULane, and report the results in Table 4. Compared

with the existing Transformer structures, our method has a big advantage of extracting features at multi-scales, which play an important role in the process of lane detection. Compared with the Deformable Transformer, our method improves the detection accuracy by 4.36%. Further, as shown in Table 4, lane features cannot be fully obtained by relying only on high-level features $p_0$ or low-level features $p_3$. The Siamese refinement structure from high to low can better integrate the global semantics information and finer-scale features, especially when roads are obscured by other vehicles or affected by poor outdoor lighting conditions.

| Transformer | Refine level | F1 score | FPS |
|---|---|---|---|
| ViT [8] | - | 72.46 | - |
| DETR [3] | - | 74.15 | - |
| Deform [42] | - | 75.65 | - |
| PriorLane [30] | - | 76.27 | - |
| **Our method** | $p_0$ | 76.87 | **210** |
| **Our method** | $p_3$ | 78.15 | 208 |
| **Our method** | $p_0 \rightarrow p_3$ | **80.01** | 172 |

Table 4: Ablation study results of the Lane Transformer on the CULane dataset with the same backbone (Swin Transformer tiny).

| Loss | F1 OpenLane | F1 CULane |
|---|---|---|
| w/o IoU | 64.28 | 77.16 |
| smooth-$L1$ | 65.52 | 77.86 |
| RIoU [22] | 67.28 | 79.01 |
| LIoU [41] | 67.75 | 79.24 |
| Curve-IoU | **68.34** | **80.01** |

Table 5: Effect of our proposed Curve-IoU on OpenLane and CULane. "w/o IoU" denotes optimizing with no IoU loss.

### 4.5.2 Curve-IoU Loss

To help the regression of lane lines, we propose the Curve-IoU loss. In Table 5, we compare our Curve-IoU with different types of IoU for lane detection. "w/o IoU" denotes optimizing the network with no IoU loss, using only classification and L1 regression loss, which only yields an F1 score of 77.16%. LIoU was proposed by CLRNet as a similarity function to calculate the distance between the prediction and ground truth. However, LIoU does not match the L1 distance when there is a long distance between the prediction lane line and ground truth. Compared with LIoU, our Curve-IoU takes the L1 distance into account, which helps the regression of lane lines with curves. On the CULane dataset, Curve-IoU achieves an F1 score of 80.01%, which is 2.85% higher than w/o IoU and 0.77% higher than LIoU. Its effect on Transformer-based lane detection is validated.

## 5 Discussion

Our method achieves state-of-the-art results on three benchmark datasets and performs the best among all the Transformer-based methods, especially in some challenging and complex road conditions. However, we still find that the current known anchor-based methods are competitive in the "Shadow", "Noline", and "Arrow" scenes on the CULane dataset, which is the opposite of the OpenLane dataset. Also, the improvement of our method is larger on the OpenLane dataset. For this interesting phenomenon, we assmue the reason behind it is that the CULane dataset contains 88K images for training while the Openlane dataset contains 160K, which is almost twice that of CULane and includes more complex and various scenes. Our Transformer-based method performs relatively better on a larger dataset compared to anchor-based and CNN-based methods, that is, our method has a much bigger potential for scaling up if the data and computation resources are sufficient. Considering the current state of affairs in vision-based autonomous driving systems, deep learning algorithms are advancing and models are expanding, yet the systems are still constrained by the size and diversity of datasets. We call for larger and better datasets featuring more diverse scenes for further research.

## 6 Conclusions

In this paper, we proposed a novel Siamese Transformer structure with hierarchical refinement, which achieves state-of-the-art results on three benchmark datasets. Specifically, we developed a high-to-low hierarchical refinement Transformer structure called LATR to refine key points of lane lines, which compensates for the Vision Transformer's deficiency in extracting finer-scale features. Also, we proposed a novel Curve-IoU loss tailored for the long and thin shape of lane lines, which helps supervise the regression of lane lines with different offsets. Extensive experiments confirmed that our model effectively handles complex scenarios, particularly when lane lines are heavily obscured by other vehicles or compromised by poor lighting conditions.

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

## A  Evaluation Metrics

For OpenLane [5] and CULane [26], we adopt the F1 score proposed by SCNN [26] as the evaluation metric. Intersection-over-Union (IoU) between the ground truth (GT) and the predicted lane line of the model is calculated to determine whether a sample is True Positive (TP), False Positive (FP), or False Negative (FN). The IoU and F1 score are calculated as in the following formulas:

$$Precision = \frac{TP}{TP + FP}, \quad Recall = \frac{TP}{TP + FN}, \tag{7}$$

$$IoU = \frac{Intersection}{Union}, \tag{8}$$

$$F1 = \frac{2 \times Precision \times Recall}{Precision + Recall}. \tag{9}$$

For Tusimple [34], the evaluation metrics are composed of three official indicators: accuracy, False Positive Rate (FPR), and False Negative Rate (FNR). The accuracy is calculated as:

$$Accuracy = \frac{\sum_{clip} C_{clip}}{\sum_{clip} S_{clip}}, \tag{10}$$

where $C_{clip}$ is the number of correct points and $S_{clip}$ is the number of ground truth (GT) points in an input image. If the accuracy of a predicted lane is greater than 85%, it will be considered a True Positive (TP). F1 score is also used in the evaluation.

## B  More Ablation Studies

### B.1  Overall Ablation Study

To verify the roles of our proposed lane Transformer (LATR) and Curve-IoU in our method, we carry out an overall ablation study with the same baseline LSTR [18]. The results of the ablation study are shown in Table 6. LATR can greatly improve the detection accuracy of lane lines, which improves the F1 scores with 8.38% and 6.45% increase on OpenLane and CULane, respectively. This is because our proposed LATR can better integrate global semantics information and finer-scale features, especially when roads are obscured by other vehicles or affected by poor outdoor lighting conditions. What's more, our proposed Curve-IoU further improves the F1 scores by 1.23% and 1.66% on OpenLane and CULane, respectively. These results show that our proposed Curve-IoU can improve the accuracy of lane detection, especially for roads with curves.

| LATR | Curve-IoU | F1 OpenLane | F1 CULane |
|------|-----------|-------------|-----------|
|      |           | 59.96       | 73.56     |
| ✓    |           | 67.11       | 78.35     |
|      | ✓         | 61.86       | 76.05     |
| ✓    | ✓         | **68.34**   | **80.01** |

Table 6: Results of the overall ablation study with the same backbone. We conduct the overall ablation study based on the same baseline LSTR [18].

| Number of Lane anchors | F1 OpenLane | F1 CULane | FPS |
|------------------------|-------------|-----------|-----|
| 50                     | 65.64       | 79.16     | **208** |
| 100                    | 67.56       | 79.75     | 195 |
| 150                    | 68.34       | 80.01     | 172 |
| 200                    | **68.36**   | 79.91     | 150 |
| 250                    | 68.18       | **80.05** | 131 |

Table 7: Results of the ablation study on the number of lane anchors. "FPS" denotes the FPS on the CULane dataset.

### B.2  Number of Lane Anchors

Our method generates a fixed number of lane anchors, much more than the maximum number of lane lines in the image. Here we conduct an ablation study on the number of lane anchors. As shown in Table 7, when we increase the number of lane anchors from 50 to 150, the accuracy of lane detection on both OpenLane and CULane is improved. This demonstrates that increasing the number of lane anchors is beneficial to the detection of lane lines. However, when we further increase the number of

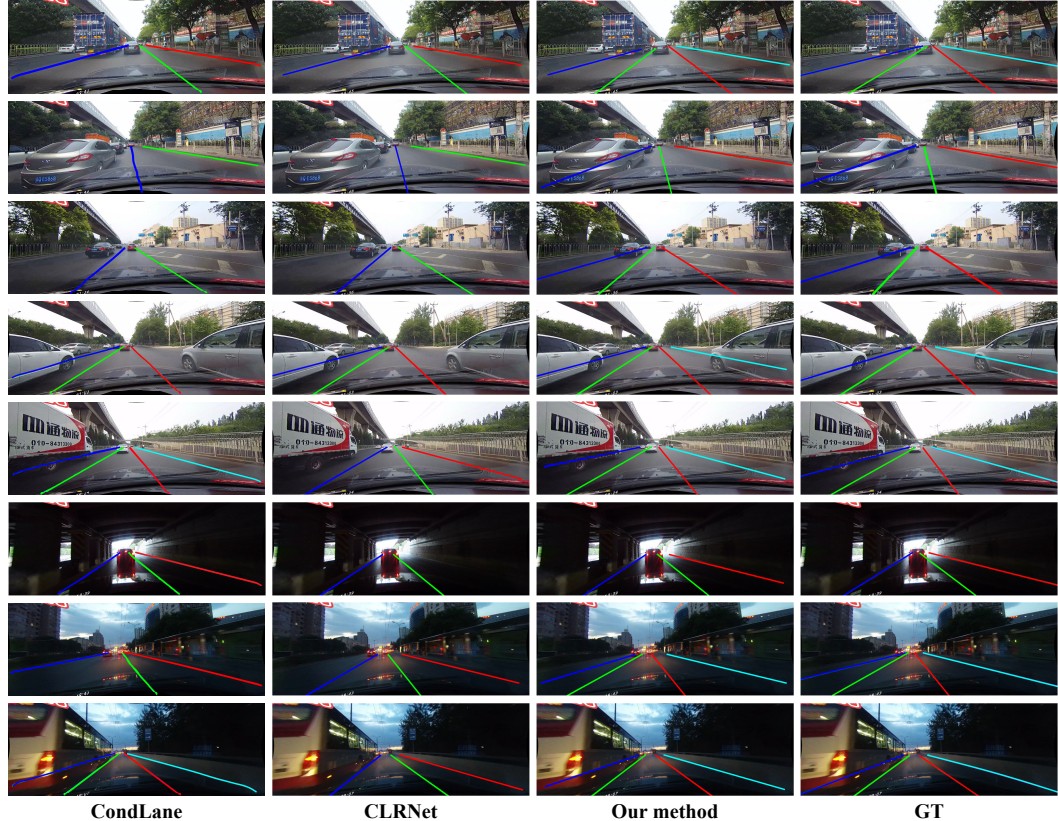

| CondLane | CLRNet | Our method | GT |

Figure 6: More visualization results of Ground Truth (GT), CondLaneNet [17] (CondLane), CLRNet [41], and our method on the CULane dataset [26]. The results of CondLaneNet and CLRNet are generated with ResNet18 and ours are generated with Swin Transformer tiny. Different lane lines are marked by different colors.

lane anchors, the enhancement is not obvious or even appears to be decreasing. This is because when the number of lane anchors reaches a certain value, adding more lane anchors will cause redundancy. 150 lane anchors are enough for the network to detect lane lines in the image.

## B.3 More Studies on Backbone

In order to evaluate the bias caused by the backbone net, we conduct ablation experiments on backbones. We find that the Swin Transformer with hierarchical features can better extract image features for subsequent processing. Therefore, we choose the Swin Transformer as our backbone network, which is an open-source backbone widely used by other works. The results show that our approach achieves better results with higher FPS and lower GFlops. We also replace the backbone of previous methods (e.g., CLRNet) with Swin Transformer and train these models under equal situations. The results are shown below in Table 8.

| Method | Backbone | F1 score on CULane |
|---|---|---|
| CondLaneNet | Swin-tiny | 77.16 |
| CondLaneNet | Swin-base | 77.84 |
| CLRNet | Swin-tiny | 79.05 |
| CLRNet | Swin-base | 79.73 |
| Ours | Swin-tiny | 80.01 |
| Ours | Swin-base | **80.85** |

Table 8: Replace the backbones of previous methods with Swin Transformer.

### B.4 Scaling up

We only use a single LATR module for each pyramid level, to find out the potential scaling-up ability of our model, we add more layers to each level, and the results are shown in Table 9 and 10. It can be seen from the results that more LATR added at the lower feature levels can improve performance, while not leading to much increase in parameters. As a result, we think it is effective to add two LATR modules at both the lowest and second-lowest feature levels.

| Number of LATR Module | F1 Score on CULane |
|---|---|
| 1 | 80.01 |
| 2 | 80.32 |
| 3 | 80.43 |

Table 9: Performance with Different Numbers of LATR Modules.

| Layer Level | F1 Score on CULane | Params |
|---|---|---|
| 1 | 80.10 | 29.186 M |
| 2 | 80.15 | 29.186 M |
| 1 & 2 | 80.22 | 29.592 M |

Table 10: Impact of Adding 2 LATR Modules to lowest and second lowest feature levels

## C    More Visualization Results

More visualization results of Ground Truth (GT), CondLaneNet [17], CLRNet [41], and our proposed method on the CULane dataset [26] are shown in Fig. 6. Compared with CondLaneNet and CLRNet, our method is more effective in detecting lane lines on congested roads and in different lighting environments, which demonstrates the robustness of our method.

