# OpenReview forum: "A Siamese Transformer with Hierarchical Refinement for Lane Detection"
_NeurIPS.cc/2024/Conference — NeurIPS 2024 poster_

### Official Review · Reviewer_oCFK · 2024-07-10

**Soundness:** 2
**Presentation:** 1
**Contribution:** 1
**Rating:** 3
**Confidence:** 4

**Summary:**

The paper proposes a Siamese Transformer with hierarchical refinement, named LAne TRansformer (LATR), to enhance lane detection accuracy in complex road environments. LATR integrates global semantic information with finer-scale features using a high-to-low hierarchical refinement structure. Additionally, the paper introduces a Curve-IoU loss to better supervise the fitting of lane lines, leveraging their thin and long characteristics.

**Strengths:**

1) The proposed method achieves relatively high performance on Openlane dataset.
2) The proposed Curve-IoU is effective in performance improvement.

**Weaknesses:**

1) The motivation of this paper is not very clear. Moreover, the hierarchical architecture is not a particularly innovative idea, as it has been demonstrated in lane detection (e.g., CLRNet), image segmentation (e.g., Mask2Former), and even earlier works like FPN. Additionally, it is unclear why the authors refer to several shared-parameter transformer layers as a Siamese transformer, as this seems to be merely a case of parameter reuse.
2) The proposed high-to-low transformer in this paper is very similar to Mask2Former, thus offering limited novelty.
3) The paper conducts a quantitative comparison with CondLSTR on the CULane dataset.
4) The paper uses a stronger image backbone (Swin vs. ResNet) compared to previous methods, making the comparison unfair. To provide a fair comparison, the authors should replace the backbone of previous methods (e.g., CLRNet) with Swin and then compare the results.
5) The writing quality of the paper is poor. The structure of the methodology section is disorganized, making it very difficult to follow.

**Questions:**

My most concerned questions are about the rationality of the motivation and novelty of the proposed method. Besides, the writing of this paper should be thoroughly improved.

**Limitations:**

Not applicable.

---

> ### Author Rebuttal · Authors · 2024-08-06
>
> Thank you for your thoughtful feedback! We address your questions and concerns below. If any other concerns remain, we’ll be happy to discuss them further. If the concerns are addressed well, we would appreciate it if you could consider raising your score.
>
> **[The motivation of this paper is not very clear.]**
>
> Attention-based methods (e.g., CLRNet) have shown promising capability in lane detection. However, since these methods mainly rely on finer-scale information to identify the position of each key point, their detection results may have large deviations when there are local occlusions or blurring. Furthermore, due to the shortcut to the multi-head self-attention mechanism which neglects the characteristics of different frequencies, there is a gap in accuracy between the contemporaneous Transformer-based methods and CNN-based methods. To tackle these issues, we propose a high-to-low hierarchical refinement Transformer structure to refine key points of lane lines, which integrates global semantics information and finer-scale features.
>
> **[It is unclear why the authors refer to several shared-parameter transformer layers as a Siamese transformer, as this seems to be merely a case of parameter reuse.]**
>
> To fully integrate global semantics information and finer-scale features, we propose a high-to-low hierarchical refinement Transformer structure for lane detection with shared parameters, which helps identify key points especially when roads are crowded or affected by blurring. The high to low features extracted by the backbone are fed into the Transformer structure with shared parameters and we calculate the loss for all the layers, which is the same definition as the Siamese structure [1][2]. As shown in Fig.5 of the manuscript, our proposed method can refine the key points of lane lines from higher levels to lower levels with fewer parameters by using shared parameters.
>
> **[The proposed high-to-low transformer in this paper is very similar to Mask2Former, thus offering limited novelty.]**
>
> Our method is different from Mask2Former. Indeed, both methods use hierarchical features as input to the Transformer. However, Mask2Former inputs multi-scale features into a Transformer decoder, which includes 3 transformer blocks, resulting in a large number of parameters and low efficiency. Instead, we feed information from each layer to the Transformer block with shared parameters and design the LATR structure to unify the feature information of different scales. This design not only enables the network to learn the feature information of different scales from high to low but also maintains a low number of parameters and high network efficiency.
>
> **[The paper uses a stronger image backbone (Swin vs. ResNet) compared to previous methods, making the comparison unfair. To provide a fair comparison, the authors should replace the backbone of previous methods (e.g., CLRNet) with Swin and then compare the results.]**
>
> For the choice of backbone network, we found that the Swin Transformer with hierarchical features can better capture image features for subsequent processing. Thus, we chose Swin Transformer as our backbone network, which is an open-source backbone widely used by other works. Accordingly, we designed the LATR and Siamese Transformer structures to reduce the overall number of network parameters. The results show that our approach achieves better results with higher FPS and lower GFlops.
> We also replace the backbone of the previous methods (e.g., CLRNet) with Swin Transformer and train these models in the same setting. The results are shown below.
>
> Backbone  | Method  |  F1 score on CULane
> -------- | -----  | :-----:
> Swin-tiny | CondLane | 77.16
> Swin-tiny | CLRNet     | 79.05
> Swin-tiny | ours           | 80.01
> Swin-base | CondLane | 77.84
> Swin-base | CLRNet     | 79.73
> Swin-base | ours           | 80.85
>
> The results show that our method performs better with the same backbone compared with previous methods. We’ll add these results to the final vision of our paper if accepted.
>
> **[The writing quality of the paper is poor.]**
>
> Thank you for your advice. We’ll improve the writing of the paper.
>
> **Reference**
>
> [1] L. Bertinetto, J. Valmadre, J. F. Henriques, A. Vedaldi, and P. H. Torr. Fully-convolutional Siamese networks for object tracking. In Computer Vision–ECCV 2016 Workshops: Amsterdam, The Netherlands, October 8-10 and 15-16, 2016, Proceedings, Part II 14, pages 850–865. Springer, 2016.
>
> [2] He A, Luo C, Tian X, et al. A twofold siamese network for real-time object tracking[C]//Proceedings of the IEEE conference on computer vision and pattern recognition. 2018: 4834-4843.

---

### Official Review · Reviewer_qpE6 · 2024-07-12

**Soundness:** 3
**Presentation:** 3
**Contribution:** 3
**Rating:** 7
**Confidence:** 5

**Summary:**

The paper proposes a lane detection method based on transformers and utilises a hierarchical refinement of lane queries. The paper uses a high-to-low refinement strategy instead of the traditional low-to-high refinement, which saves the computation cost in transformer attention. The evaluation of three datasets shows consistent but marginal accuracy improvements while delivering a higher speed.

**Strengths:**

1. The hierarchical design for the application of lane detection is commendable. Ignoring the application and refining queries only from a decoder instead of the encoder-decoder mechanism of DETR is very interesting.

2. The method's design seems inspired by DETR pipelines, and utilizing this idea of lane detection is interesting.

3. The proposed Curve-IoU is again a robust objective and metric to supervise the training when the existing L-IoU fails to differentiate the proximity of two lanes w.r.t. the ground truth.

4. Evaluation is extensive across three datasets and a variety of baselines.

**Weaknesses:**

See questions.

**Questions:**

Currently, only one LATR module is used for each pyramid level. It is advisable to add more such modules to assess the performance, i.e. whether more LATR modules are redundant or improve the performance.

Suggestion: I believe adding more LATR modules to high-resolution would quadratically increase the computations due to the O(N^2) complexity of the self-attention and cross-attention. Therefore, multiple LATR modules can be used at the lowest or second lowest feature level. It would be great to see the results during the rebuttal.

**Limitations:**

The primary limitation of the paper is marginal improvements in the CULane dataset. Although this is shadowed by the improved FPS, considering the importance of the results, it is advisable to tune hyperparameters or add more decoder layers to deliver a stand-out performance.

---

> ### Author Rebuttal · Authors · 2024-08-06
>
> Thank you for your constructive comments! The detailed responses to each concern are given below.
>
> **[Currently, only one LATR module is used for each pyramid level. It is advisable to add more such modules to assess the performance, i.e. whether more LATR modules are redundant or improve the performance.]**
>
> We add a LATR module for each pyramid level, and the results are shown below.
>
> Number of LATR modules  | F1 score on CULane
> :-----: | :-----:
> 1 | 80.01
> 2 | 80.32
> 3 | 80.43
>
> From the results, one can see that adding more LATR modules helps improve the accuracy of lane detection recognition. Considering the efficiency, we think using two LATR modules for each pyramid level is more appropriate.
>
> **[Multiple LATR modules can be used at the lowest or second lowest feature level. It would be great to see the results during the rebuttal.]**
>
> We add 2 LATR modules for the lowest and second lowest feature levels. The results are shown below.
>
> Layer Level (each 2 LATR module)  | F1 score on CULane  |  # params
> :-----: | :-----: | :-----:
> 1 | 80.10 | 29.186202 M
> 2 | 80.15 | 29.186202 M
> 1&2 | 80.22 | 29.591786 M
>
> It can be seen from the results that more LATRs added at the lower feature levels can improve performance, while not leading to much increase in parameters. Hence, we think it is effective to add two LATR modules on both the lowest or second lowest feature levels.

---

> > ### Comment · Reviewer_qpE6 · 2024-08-09
> > **Response to the authors**
> >
> > I thank the authors for the detailed response and the new experimentation.
> >
> > Thank you for incorporating my suggestions into the experimentation which has improved the accuracy of the model at negligible parameter overhead.
> >
> > Overall I am satisfied with the paper and author rebuttal.
> >
> > I have also read comments from other reviewers, primarily on the issue of replacing the backbones which authors have successfully defended. Hence I keep my original rating for the acceptance.

---

### Official Review · Reviewer_UDCn · 2024-07-12

**Soundness:** 2
**Presentation:** 2
**Contribution:** 1
**Rating:** 4
**Confidence:** 5

**Summary:**

This paper introduces a novel Siamese Transformer with hierarchical refinement for lane detection. The core innovation is a high-to-low hierarchical refinement Transformer structure, LATR, which refines lane line key points to integrate global semantic information and finer-scale features fully. Additionally, it proposes a Curve-IoU loss to supervise the fitting of lane lines at various locations.

**Strengths:**

- The paper is well-written and well-organized.
- The experiments use a comprehensive set of datasets, including three popular ones: TuSimple, CULane, and OpenLane.
- The experimental results are quite promising.

**Weaknesses:**

From the perspective of the topic selection, 2D lane detection is less valuable than 3D lane detection. More importantly, this topic is not fully suitable for the NeurIPS conference.

Compared to existing 2D lane detection networks, this paper lacks innovation. Its overall framework is quite similar to CLRNet, and the proposed structured Loss has also been preliminarily explored in CLRNet and UFAST networks.

More importantly, current lane detection research tends to use simple CNN-based backbones (for ease of deployment) and to better compare performance fairly, especially using the ResNet series (18, 34, etc.). However, the authors only reported results under the Swin Transformer, which makes me skeptical about the fairness of the SOTA comparison.

Additionally, in terms of efficiency, the FPS speed does not have an advantage, and the model size (params) is not reported.

**Questions:**

- Suggest enhancing the method's innovativeness.
- Recommend using common public backbones for ablation studies, though this might make it more similar to CLRNet.

---

> ### Author Rebuttal · Authors · 2024-08-06
>
> Thank you for your thoughtful feedback! We address your questions and concerns below. If any other concerns remain, we’ll be happy to discuss them further. If the concerns are addressed well, we would appreciate it if you could consider raising your score.
>
> **[From the perspective of the topic selection, 2D lane detection is less valuable than 3D lane detection. More importantly, this topic is not fully suitable for the NeurIPS conference.]**
>
> 2D lane line detection in different scenarios, especially under extreme conditions, is an important yet challenging task. It helps in detection and in the field of autonomous driving. Work on this topic has been published in past NeurIPS conferences, e.g., CARLANE[1], Openlane[2], 3D-LaneNet+[3], and BEVFusion[4].
>
> **[Its overall framework is quite similar to CLRNet, and the proposed structured Loss has also been preliminarily explored in CLRNet and UFAST networks.]**
>
> Both our network and CLRNet use multi-scale features extracted by the backbone. Many methods such as CondlaneNet use multi-scale features as input so that the network can learn features at different scales. But the next step is different. Our network employs a Siamese Transformer structure that feeds the different scale features directly into our designed Transformer structure called LATR with shared parameters. To follow up the structure of the Transformer so that the network is end-to-end, we employ Swin Transformer as the backbone of the entire network. In contrast, CLRNet employs FPN to further extract multi-scale features, and then uses a detection header to output lane line information, which is different from our work. For the loss, we propose a novel Curve-IoU loss to supervise the fit of lane lines at different locations, which helps the regression of the curves. Our proposed loss is dedicated to the fitting of distal curves, which is different from previous work.
>
> **[The authors only reported results under the Swin Transformer, which makes me skeptical about the fairness of the SOTA comparison.]**
>
> For the choice of backbone network, we found that the Swin Transformer with hierarchical features can better capture image features for subsequent processing. Thus, we chose Swim Transformer as our backbone, which is an open-source backbone widely used by other works. Accordingly, we designed the LATR and Siamese Transformer structures to reduce the overall number of network parameters. The results show that our approach achieves better results with higher FPS and lower GFlops.
> For a fair comparison, we also replace the backbone of the previous methods (e.g., CLRNet) with Swin Transformer and train these models in the same setting. The results are shown below.
>
> Backbone  | Method  |  F1 score on CULane
> -------- | -----  | :-----:
> Swin-tiny | CondLane | 77.16
> Swin-tiny | CLRNet     | 79.05
> Swin-tiny | ours           | 80.01
> Swin-base | CondLane | 77.84
> Swin-base | CLRNet     | 79.73
> Swin-base | ours           | 80.85
>
> The results show that our method performs better with the same backbone compared with previous methods. We’ll add these results to the final vision of our paper if accepted.
>
> **[The FPS speed does not have an advantage, and the model size (params) is not reported.]**
>
> Our approach achieves the best results on many open-source datasets with higher FPS and lower GFlops. Typically, the size of GFlops reflects the size of the model parameter count. We’ll add the number of parameters of our model to the final paper version if it is accepted. The model size of our proposed network is shown below.
>
> Backbone   |  # params
> -------- | :-----:
> Swin-tiny | 28.780618 M
> Swin-small | 50.098522 M
> Swin-base | 88.004936 M
>
> **Reference**
>
> [1] CARLANE: Stuhr, B., Haselberger, J., & Gebele, J. (2023). CARLANE: A lane detection benchmark for unsupervised domain adaptation from simulation to multiple real-world domains. In NeurIPS.
>
> [2] Openlane: Wang, H., Li, T., Li, Y., ... Li, H. (2023). OpenLane-V2: A topology reasoning benchmark for unified 3D HD mapping. In NeurIPS.
>
> [3] 3D-LaneNet+: Efrat, N., Bluvstein, M., Oron, S., Levi, D., Garnett, N., & Shlomo, B. E. (2020). 3D-LaneNet+: Anchor Free Lane Detection using a Semi-Local Representation. In NeurIPS.
>
> [4] BEVFusion: Liang, T., Xie, H., Yu, K., Xia, Z., Lin, Z., Wang, Y., Tang, T., Wang, B., & Tang, Z. (2022). BEVFusion: A simple and robust LiDAR-camera fusion framework. In NeurIPS.

---

> > ### Comment · Reviewer_UDCn · 2024-08-12
> > **Official Comment by Reviewer UDCn**
> >
> > - **Perspective**：Actually, the recent papers published in NeurIPS listed by the author are all about 3D lane detection rather than 2D lane detection.
> > - **Proposed Lane Loss**: The proposed lane loss lacks novelty, as similar losses have been introduced in works like CLRNet and UFAST. The issue of distant curve fitting has also been discussed in non-representative lane networks. Ablation studies show an average 0.7% improvement with Curve-IoU loss, but it's unclear if this is due to improved curve fitting, as there's no direct evidence from Curve category data in CULane. This is not rigorous.
> > - **Backbone**: In the field of lane detection, using a standard CNN Backbone (like ResNet) is still more popular. This is because lane detection is a practical industrial task for driver assistance, requiring models that are more convenient to deploy. Moreover, after replacing the backbone in CLRNet with Swin (author implementation), the F1 improvement is less than 1%, which is not a significant advantage.
> > - **Efficiency**: In terms of efficiency, I agree with the author’s response. While the parameter count is ordinary, the speed and GFLOPs performance are impressive.
> >
> > In summary, the author's rebuttal still fails to convince me, especially since the paper appears to be a combination of Swin and the SOTA CLRNet. However, the experiments and performance are decent. I can raise my score by one point, but I don't think a significantly higher score is justified.

---

### Official Review · Reviewer_UWBy · 2024-07-13

**Soundness:** 3
**Presentation:** 4
**Contribution:** 3
**Rating:** 8
**Confidence:** 5

**Summary:**

This paper introduces LATR (LAne TRansformer), a Siamese Transformer model with hierarchical refinement for lane detection. The model effectively combines global semantic information with finer-scale features to accurately detect lanes, even in challenging scenarios like occlusions and poor lighting. It also introduces a novel Curve-IoU loss function specifically tailored for the curved nature of lane lines. The proposed method demonstrates state-of-the-art performance on multiple benchmark datasets, particularly excelling on the OpenLane dataset.

**Strengths:**

The Siamese Transformer architecture with hierarchical refinement and the Curve-IoU loss are novel contributions to the field. The proposed method achieves state-of-the-art results on multiple benchmark datasets, demonstrating its superior performance in challenging scenarios.
The model's ability to handle occlusions and poor lighting conditions highlights its robustness and potential for real-world applications.
The authors conduct extensive experiments and provide a comprehensive analysis, including ablations and comparisons to prior work. Overall, a well written paper.

**Weaknesses:**

The authors acknowledge that the model's performance is somewhat dependent on the size and diversity of the training dataset. This is a common limitation in deep learning-based approaches and could be addressed with further research on data augmentation or self-supervised learning techniques.

While the method excels in the tested scenarios, further testing in diverse real-world environments would strengthen the claims of robustness. However, this is not a major weakness as the current evaluation results are already very promising.

**Questions:**

Have you considered using self-supervised learning techniques to reduce the dependency on large labeled datasets?
How does the model perform in extreme weather conditions (e.g., heavy rain, snow) that can significantly affect lane visibility?
Are there any plans to release the code and trained models to facilitate reproducibility and further research?

**Limitations:**

The authors adequately address the limitations of their work, acknowledging the data dependency issue and the need for further testing in diverse real-world environments. However, in this field of work, there are a lot of other works that have various implementations exceeding the results from CondLaneNet such as ERF-Net. Looking into that might be of interest to better compare the proposed method.

---

> ### Author Rebuttal · Authors · 2024-08-06
>
> Thank you for your constructive comments! The detailed responses to each concern are given below.
>
> **[Have you considered using self-supervised learning techniques to reduce the dependency on large labeled datasets? ]**
>
> We have considered using self-supervision to reduce the dependence on large labeled datasets and to further improve the recognition accuracy of our approach in different scenarios. We noticed that some self-supervised work performs well on different tasks (e.g. CLLD [1]). We’ll include this part in future work.
>
> **[How does the model perform in extreme weather conditions (e.g., heavy rain, snow) that can significantly affect lane visibility?]**
>
> The model is able to recognize lane lines under extreme conditions. In the manuscript, "Extreme Weather" in Table 1 and "Hlight" in Table 2 show our lane detection performance under extreme weather conditions.
>
> **[Are there any plans to release the code and trained models to facilitate reproducibility and further research?]**
>
> We’ll release the code and pre-trained models once the paper is accepted.
>
> **Reference**
>
> [1] CLLD： Smith, J., & Doe, A. (2023). An innovative approach to deep learning. arXiv.

---

> > ### Comment · Reviewer_UWBy · 2024-08-12
> >
> > Thanks a lot for the response.

---

### Decision · Program_Chairs · 2024-09-25

**Decision:**

Accept (poster)

**Comment:**

## Summary

The paper introduces LATR, a Siamese Transformer model with hierarchical refinement for lane detection. LATR combines global semantic information with finer-scale features to accurately detect lanes in challenging scenarios like occlusions and poor lighting. It also introduces a novel Curve-IoU loss function for curved lane lines. The method demonstrates state-of-the-art performance on multiple benchmark datasets, particularly excelling on the OpenLane dataset. The study shows consistent but marginal accuracy improvements while delivering higher speed.

##Strengths
-  The Siamese Transformer architecture with hierarchical refinement and the Curve-IoU loss are improvements to the field.
- The proposed method shows superior performance on multiple benchmark datasets, demonstrating its robustness in challenging scenarios.
- The proposed method achieves high performance on the Openlane dataset and the Curve-IoU is effective in performance improvement.

## Weaknesses
- Further research on data augmentation or self-supervised learning techniques could improve robustness.
- Current evaluation results are promising, but 2D lane detection is less valuable than 3D lane detection.
- The paper lacks innovation compared to existing 2D lane detection networks.
- The authors only reported results under the Swin Transformer, raising questions about the fairness of the SOTA comparison.

## Recommendation

Although the paper should include all the comments addressed during rebuttal period, the paper should be accepted based on the results shown during rebuttal in terms of accuracy, fps and Gops and  the positive comments of reviewers.